# The Accuracy of Transurethral Bladder Resection in Detecting Bladder Cancer Histological Variants and Their Prognostic Value at Radical Cystectomy

**DOI:** 10.3390/jcm11030550

**Published:** 2022-01-22

**Authors:** Giovanni La Croce, Richard Naspro, Marco Finati, Federico Pellucchi, Mario Sodano, Michele Manica, Michele Catellani, Andrea Gianatti, Marco Roscigno, Luigi Filippo Da Pozzo

**Affiliations:** 1Department of Urology, ASST Papa Giovanni XXIII, 24127 Bergamo, Italy; giovanni.lacroce@gmail.com (G.L.C.); nasprorichard@gmail.com (R.N.); m.finati@outlook.it (M.F.); pellucchi@yahoo.it (F.P.); mariosodano86@gmail.com (M.S.); manicaxmichele@gmail.com (M.M.); mcatellani@asst-pg23.it (M.C.); lfdapozzo@gmail.com (L.F.D.P.); 2Department of Pathology, ASST Papa Giovanni XXIII, 24127 Bergamo, Italy; agianatti@asst-pg23.it; 3School of Medicine, University of Milano-Bicocca, 20126 Milan, Italy

**Keywords:** bladder cancer, histological variants, transurethral resection

## Abstract

Objectives: to investigate the accuracy of transurethral resection of bladder tumours (TURBT) in detecting histological variants (BHV) at radical cystectomy (RC) and to evaluate the impact of TURBT before cystectomy on oncological outcomes. Methods: Data of 410 consecutive RCs were assessed. Positive and negative predictive values were used to assess the accuracy of TURBT in detecting BHV. Cohen’s Kappa coefficient was used to calculate the agreement grade. Logistic regression analysis predicted features based on the presence of BHV at TURBT. Multivariable backward conditional Cox regression analysis was used to estimate oncological outcomes. Results: A total of 73 patients (17.8%) showed BHV at TURBT as compared to 108 (26.3%) at RC. A moderate agreement in histological diagnosis was found between TURBT and RC (0.58). However, sensitivity and specificity in detecting BHV were 56% and 96%, respectively. Furthermore, positive predictive value (PPV) was 84.7% and negative predictive value (NPV) was 84.6%. Presence of BHV at TURBT was an independent predictor for pathologic upstage, albeit not a predictor for positive nodes or positive surgical margins. However, at multivariable analysis adjusted for all confounders, presence of BHV at TURBT was an independent predictor for recurrence after RC, but not for survival. Conversely, the presence of BHV at RC was an independent predictor for both recurrence and survival. Conclusion: There was a moderate agreement between TURBT and RC histopathological findings. TURBT, alone, could not provide an accurate and definitive histological diagnosis. Detection of BHV in TURBT specimens is not an independent predictor of oncological outcomes; indeed, only pathological features at RC are associated with worse survival. However, BHV presence in cystectomy specimens resulted as an independent predictor of both cancer-specific and overall mortality.

## 1. Introduction

Urothelial carcinoma of the bladder (UCB) may exhibit a wide range of histological differentiation besides the conventional pure UCB [1]. Specifically, the combination of UCB with other differentiated patterns is described in up to 25% of patients who underwent a transurethral resection of bladder tumour (TURBT), with an increase of up to 30% when considering radical cystectomy (RC) specimens [2,3].

While the association of bladder urothelial histological variants (BHV) with adverse clinical–pathological features and even worse outcomes after surgery have been widely proven [4], its impact on neoadjuvant complementary therapies is still under debate, since these patients are often excluded from clinical trials. However, preliminary results have shown a possible role of BHV as a driver to in response both to neoadjuvant chemotherapy (NAC) [5] and immunotherapy [6]. As a result of these studies, BHV could be a possible discriminant for neoadjuvant treatment versus direct RC.

In this perspective, an early detection of BHV must be pursued to optimize the management, the treatment option and follow-up of the disease. For this purpose, the World Health Organization 2016 classification for bladder tumours recommend an accurate morphological description of all specimens, both from TURBT and RC [7]. We evaluated the reliability of TURBT in detecting BHV in a cohort of patients who underwent RC from a single-tertiary referral centre and the impact of findings at TURBT on survival and recurrence of patients subsequently treated with RC.

## 2. Materials and Methods

### 2.1. Patient Selection

The study was conducted in accordance with the declaration of Helsinki ethical principles and good clinical practices and was approved by the local Ethical Committee (number 160/19). We retrospectively collected data from 410 consecutive patients treated with RC from October 2008 to September 2019 at the Department of Urology, Papa Giovanni XXIII Hospital, Bergamo. In our department, approximately 60 open radical cystectomies are performed per year. Surgical technique was standardized, and extended lymph node dissection was performed at the time of surgery, as previously described [1]. Moreover, in the same department, approximately 500 TURBT procedures are performed every year by 8 different surgeons using a standardized good quality approach according to the European Association of Urology (EAU) Guidelines [8].

After discharge, all patients signed an informed consent and were enlisted in a standardized follow-up regimen, which consisted of half-yearly visits for two years with laboratory tests, chest–abdomen–pelvis CT scan and urine cytology. After this period, patients could decide to continue yearly follow-ups at our department or with their general practitioner. Our institutional cystectomy database was frozen and examined in March 2020 for current analysis.

Eleven patients (2.7%) who underwent NAC were excluded from the analysis, as they represented a very small but possible confounder for outcome analysis.

Pathological evaluation and classification:

All surgical specimens were processed, described and reviewed by a team of dedicated uropathologists. For urothelial cancer, grade was classified according to the WHO/ISUP 2016 grading system [7]. Pathological stage was assigned following the current American Joint Committee on Cancer 2017 TNM staging system (VIII edition) [9].

Lymph vascular invasion (LVI) was defined as invasion of cancer cells within an endothelium-lined space without underlying muscular walls [10,11,12]. A macroscopic (R2) or microscopic (R1) positive surgical margin was defined as a presence of a tumour located in inked areas of tissue on the RC specimen [11]. BHV was defined as the combination of pure UCB with other differentiated histotypes. Pattern of classification was based predominantly on morphological features on haematoxylin- and eosin-coloured sections, following the WHO 2004 (third edition) classification for tumours of the urothelial tract [12]. The threshold used for variants was 5%, and no cases were described as having more than one variant in the same specimen. In particular, we analysed the presence of BHV at transurethral resection and matched it with histological findings at cystectomy to evaluate any correlation with the overall outcomes. Patients with no evidence of cancer at RC (pT0) were excluded from the correlation analysis.

### 2.2. Outcome Measure

The primary endpoint was to evaluate the reliability of TURBT in detecting histological variants.

Secondly, we evaluated the impact of the presence of BHV on oncological outcomes (recurrence-free survival, cancer-specific mortality (CSM) and overall mortality (OM)). Recurrence-free survival was calculated from time of surgery to recurrence, while CSM and OM were from the time of diagnosis to metastases or death, accordingly. Recurrence was defined as local when cancer reoccurred in the operative site and regional when disease reoccurred in the field of lymph node dissection template at cystectomy, [13] whilst distant metastasis is considered recurrence in lymph nodes out of the template field or in other organs. Finally, we evaluated the presence of possible preoperative prognostic factors at TURBT of patients with BHV at RC.

### 2.3. Statistical Analysis

Clinical–pathological characteristics of our population were studied using descriptive statistics. Median and interquartile range (IQR) were reported for age at surgery and range for postoperative follow-up. TURBT accuracy in detecting BHV was measured in terms of sensitivity, specificity and positive and negative predictive values (VPP, VPN). The Cohen’s Kappa coefficient assessed the concordance of presence of BHV between TURBT and RC. Values range between 0 and 1 (0–0.2 poor agreement, 0.21–0.40 fair agreement, 0.41–0.60 moderate agreement, 0.61–0.8 good agreement, 0.81–1 very good agreement). We did not consider for correlation analysis 31 patients without residual tumour at RC (pT0). Univariable and multivariable logistic regression analysis investigated predictors of adverse pathological features at RC. Univariable and multivariable backward conditional Cox regression analysis addressed overall mortality (OM), cancer-specific mortality (CSM) and recurrence, adjusted for all available confounders, on the overall population. Kaplan–Meier (KM) survival curves evaluated recurrence-free survival, CSM- and OM- free rates between patients with BHV at TURBT and those with pure UCB and in the overall population. All tests were two-sided, and statistical significance was defined as *p*-value <0.05. All analyses were performed using SPSS Statistics© 22 (SPSS, IBM Corporation, Armonk, NY, USA).

## 3. Results

Median age at RC was 71 years (IQR 40–90); 340 patients were males (82.9%) and 70 females (17.1%). The median time elapsed between radical surgery and first recurrence in those patients who experienced recurrence (155/410; 37.8%) was 9 months (IQR 4–15), while the median time to death for those patients who died (190/410; 46.3%) was 19 months (IQR 11–42). Postoperative characteristics of RC patients are reported in Table 1. Table 2 describes histology comparing TURBT and RC specimens. A further 47 patients (12.4%) with pure UCB at TURBT showed BHV at subsequent RC, specifically, thirty-one UCB with squamous differentiation (8.1%), six micropapillary (1.5%), four UCB with glandular differentiation (1.0%), two nested (0.5%), two giant cell (0.5%), one sarcomatoid and one small cell (0.2%).

Conversely, 11 patients with BHV (2.9%) at TURBT, in detail, 9 UCB with squamous differentiation (23.7%), 1 micropapillary (2.6%) and 1 UCB with glandular differentiation (2.6%), did not show a differentiation at the following cystectomy. In all these cases, precystectomy TURBT described a complete and radical resection. Only one patient with a microcystic tumour at TURBT was pT0 at RC.

### 3.1. Reliability of TURBT in Detecting BHV

When compared to RC, TURBT had a sensitivity and specificity in detecting BHV of 56.4% and 95.9% with 95% Confidence Interval (CI) of 46–66% and 93–98%, respectively. Positive predictive value (PPV) was 84.7% (95%CI 75–91%), and negative predictive value (NPV) was 84.7% (95%CI 80–88%). The Cohen’s Kappa coefficient showed a moderate agreement between TURBT and RC (0.58, *p*-value < 0.001).

At univariable analysis, the presence of BHV at TURBT was associated with pathologic upstage (T > 2, *p* = 0.020) and the presence of nodal metastases (*p* < 0.001) but not with positive surgical margins at RC (Table 3A–C). However, after accounting for other clinical–pathological confounders, BHV resulted as an independent predictor of extra-vesical disease only (Table 3A–C).

#### Survival Outcomes

At the time of analysis, 57% (66 out of 116) and 47% (133 out of 284) of patients with BHV and UCB were deceased, respectively.

Cancer-related death was documented in 58 (50%) with BHV and 93 (33%) with pure UCB. A total of 62 out of 116 (53%) with BHV and 107 out of 284 (38%) with pure UCB exhibited a recurrence during their follow-up.

The Kaplan–Meier curves showed the worst oncological features in patients with BHV at TURBT with a reduced survival and recurrence-free rates when compared with pure UCB (Figure 1A–C). Oncological outcomes in patients with BHV only at TURBT compared to those who presented BHV at RC (Figure 2A–C) were not statistically different, maybe due to the small number of cases.

After multivariable analysis adjusted for all confounders, the presence of BHV at TURBT was an independent predictor for recurrence but not for CSM and OM. Conversely, the presence of BHV at RC was an independent predictor for both recurrence and survival outcomes.

Moreover, multivariable Cox analyses confirmed the importance of pathological findings: extravesical disease at RC, positive surgical margins, LVI and lymph node involvement are independently associated with CSM and OM. The only histological finding at TURBT that seems to be an independent predictor of oncological outcomes is the presence of muscle-invasive cancer at TURBT (Table 4).

## 4. Discussion

Identifying BHV at TURBT is crucial for the correct management of bladder cancer patients. Indeed, the incidence rate of BHV is constantly increasing over the years, and the latest data from the WHO/ISUP society highlight an incidence of BHV of about 30% at RC.

An early recognition of BHV at TURBT is of significance to both pathologists and clinicians for several reasons. First, it is important for the pathologist to distinguish between a histological urothelial variant from a nonurothelial form, as they often present similar patterns, accounting for potential diagnostic misinterpretations. Secondly, some BHV have a different response to NAC and require different treatment modalities compared to pure UCB. Lastly, the presence of BHV may drive towards early radical surgery treatment because of the potentially worse prognosis also in those patients with BHV with non-muscle-invasive bladder cancer.

We retrospectively analysed data from patients treated with TUR who underwent RC to evaluate if diagnosis of BHV can be accurately made with TUR, alone, and to evaluate the prognostic role of BHV detection both at TURBT and RC. Those patients treated with NAC were excluded from the analysis. Our analysis calculated oncological outcomes from the date of surgery in order to make data more homogenous, as calculating from the time of diagnosis could add another interpretation bias.

On the other hand, the diagnosis of BHV at TURBT may be challenging for the pathologist due to many factors: (1) extension and proportion of the BHV of the whole tumour, (2) the pathologist’s experience and (3) possible artefacts caused by resection technique, sampling or staining [14,15,16].

Despite the efforts in improving the accuracy of both TURBT and bladder biopsies, our study confirms how these procedures could not yet provide an accurate and definitive histological diagnosis. In our study, the concordance between diagnosis of BHV at TURBT and RC is of 56%, with a sensitivity and specificity in detecting BHV of 56% and 95.9% with 95% Confidence Interval (CI) of 46–66% and 93–98%, respectively. Furthermore, positive PPV was 84.7% (95%CI 75–91%), and NPV was 84.7% (95%CI 80–88%). The Cohen’s Kappa coefficient showed a moderate agreement between TURBT and RC (0.58, *p*-value < 0.001). PPV, NPV, sensibility and specificity were calculated using as a positive test the presence of BHV at TURBT, as a negative test the presence of UCB at TURBT and as a positive disease test the presence of BHV at the cystectomy. The sensitivity of TURBT in detecting BHV is extremely variable. Previously in literature, the highest rate was observed by Ge et al., with an overall sensitivity of bladder biopsy or TURBT to detect histological variant differentiation up to 50% [17]. Our data showing a higher sensitivity are encouraging, compared to previous studies, and a very high specificity shows how an experienced and dedicated team of uropathologists could change the diagnostic accuracy. Several studies have evaluated the rate of concordance between the resection and the final pathologic examination with no satisfactory results [18,19]. Moschini et al. showed how the rate of agreement ranges between 0.11 and 0.61, considering each single variant [20]. Similarly in our cohort, the global rate of agreement, calculated with Cohen’s Kappa coefficient, was 0.61, which means a good agreement between procedures. However, 34% of BHV patients at RC in our study had been classified as UCB at a previous resection.

Indeed, from the pathological point of view, in a cystectomy specimen, BHV differentiations can be more easily identified, as the pathologist can examine more neoplastic tissue, especially for poorly differentiated neoplasms. The main discordant results between histological findings in TURBT and cystectomy were in patients with the squamous variant. The criteria for the determination of squamous differentiation are exclusively morphological, and so far there are no indications for routinely using immunohistochemical markers of squamous differentiation: urothelial carcinoma with squamous differentiation is defined by the presence of clear-cut intercellular bridges, intracellular keratinization or both or the formation of squamous pearls in a background of invasive urothelial carcinoma. These criteria are difficult to apply, especially in poorly differentiated urothelial neoplasms for which in the TURV samples there can be an underestimation of the squamous component, because the extent of squamous differentiation in urothelial carcinoma may vary and be very limited in the TURV samples.

Conversely, in a cystectomy specimen, squamous differentiation can be more easily identified, as the pathologist can examine more neoplastic tissue and more carefully look for even the slightest signs of squamous differentiation.

In this context, the importance of the role of a good quality TURBT is emerging, firstly to allow pathologists to make an accurate and detailed diagnosis immediately at TUR.

Patients with BHV at TURBT showed adverse clinical and pathologic features compared to those with pure UCB. Our analysis was concordant with previous studies [14,15]. We found that the presence of BHV at TURBT was associated with pathologic upstage, albeit not being a predictor for positive surgical margins and, unlike the paper from Abufaraj et al., we could not demonstrate that the presence of BHV at TURBT could be a predictor of nodal metastasis, even if a higher detection rate of BHV was present in our series (17.8% compared to 11%) [21].

On the contrary, BHV at TURBT was statistically associated with a higher risk of facing a recurrence after RC and with worse survival, as reported in the K–M curves. However, at multivariable analysis adjusted for all confounders, presence of BHV at TURBT was an independent predictor for recurrence but not for survival. Conversely, the presence of BHV at RC was an independent predictor for both recurrence and survival.

In this context, it should be highlighted that the moderate agreement between BHV detection at TURBT and RC indicates that TURBT, alone, could not provide an accurate and definitive histological diagnosis. Up until now, even though the detection of BHV at TURBT may suggest the possibility of a more extended disease, only the pathological stage at RC allows prediction of survival outcomes.

Nonetheless, we would like to underline the need of early detection of BHV at TURBT as a possible tool for a multimodal approach tailored to a single patient.

BHV could become a clinical tool for early RC in patients at high risk for adverse disease. For example, patients with micropapillary pattern are less sensitive to immunotherapy or NAC, to the point that one leading group has already suggested early RC should be taken into consideration in the clinical management of those patients with micropapillary histology for Ta and T1 tumours as well as for squamous cell carcinoma or sarcomatoid variant because of their aggressive behaviour with high rates of local progression and distant metastasis [16]. Additionally, the level of evidence is still not sufficient to make any recommendation at the clinical level. Lastly, pretreatment identification of the nonresponders to NAC could avoid unnecessary side effects and delay in surgery, but no validated instruments currently exist [5,6]. Our study is not devoid of limitations. First, it was a retrospective nonrandomized study of a single-centre series with an overall sample size relatively limited. Single-histotype analysis was not performed using Cohen’s Kappa coefficient due to small number of cases for certain subgroups. At the same time, when considering percentage of correlation between TURBT and RC in our study, the best agreement was found for throphoblastic, small cell carcinoma and sarcomatoid variants, followed by micropapillary and nested variants.

These subgroups have generated an extreme variability in results from different centres, limiting possible comparison among the studies. Even in the multicentric study of Cai et al. that shows a low concordance between TURBT and RC and less favourable outcomes for BHV, there were unsatisfactory results despite the study having involved 17 tertiary-referral centres [18]. In this perspective, a more accurate standardization of the description of BHV in the specimen would be desirable to guarantee a standardized pathologic review. A multicentric prospective study could partially help to confirm our findings. Another pitfall could be the absence of percentage of variant in each sample. Therefore, we assumed that any histological component would equally impact outcomes, regardless of the percentage. Indeed, it would be interesting to evaluate the impact of the percentage of histological variant in specimen. However, even if a 5% threshold was generally considered to report the presence of a histological variant, we are unable to provide the exact percentage for each patient. Compared to other similar experiences, in which the impact of BHV on survival was evaluated, in particular when analysing patients from the Eastern Mediterranean region, survival rates are similar to ours, even if our population was older (50 vs. 71 years) and with worse pathological features (≥pT3 = 25% vs. 39%) at diagnosis [22].

Conversely, the paper does present some strong points. In particular, the cohort is a faithful representative subset of a population referring to a tertiary hospital, with no previous selection of patients based on clinical–pathological characteristics. Cases were discussed in a multidisciplinary fashion, and all specimens were examined by a team of dedicated uropathologists who analysed both TURBT and respective RC specimens. This last point is maybe one of the reasons of the higher sensibility and concordance when our study is compared to others in the same topic. Furthermore, as patients who underwent NAC were excluded, potential confounders were removed from data analysis, reducing misdiagnosis factors. Simultaneously, the whole cohort underwent the same standardized follow-up schedule up to three years from surgery, regardless of clinical stage.

## 5. Conclusions

Our study demonstrated a moderate agreement between TURBT and RC histopathological findings. The detection rate of BHV at TURBT in our series was comparable with that found in literature and confirmed that endoscopic resection cannot be considered a safe and definitive diagnostic tool. However, BHV presence in cystectomy specimens resulted as an independent predictor of CSM- and OM- free rate. Considering its prognostic implications, an early detection of BHV at TURBT could be useful to drive the best disease management and offer different treatment modalities compared to pure UCB.

## Figures and Tables

**Figure 1 jcm-11-00550-f001:**
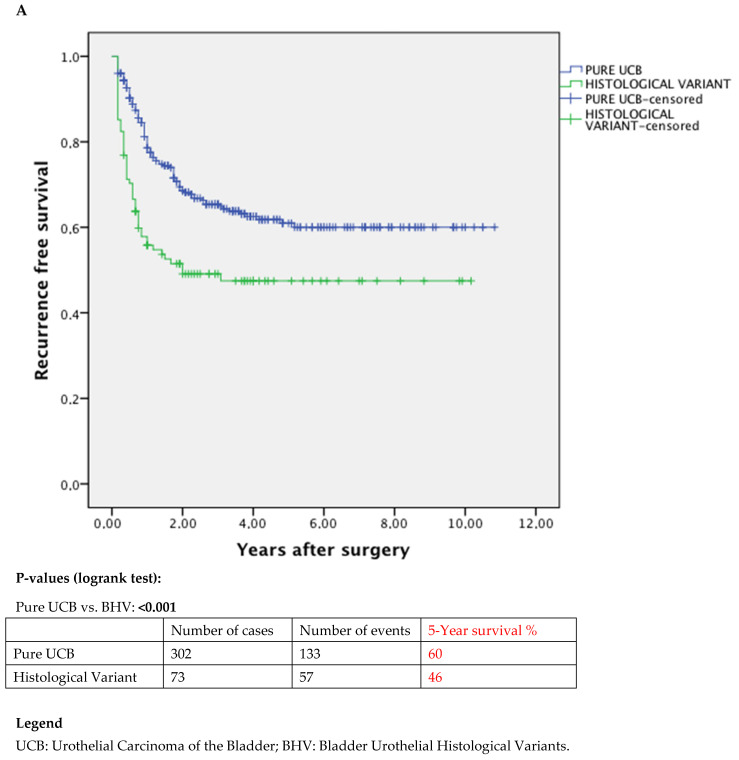
Kaplan–Meier analysis assessing recurrence-free survival (**A**), CSM-free rates (**B**) and OM-free rates (**C**) after RC, stratified by histology.

**Figure 2 jcm-11-00550-f002:**
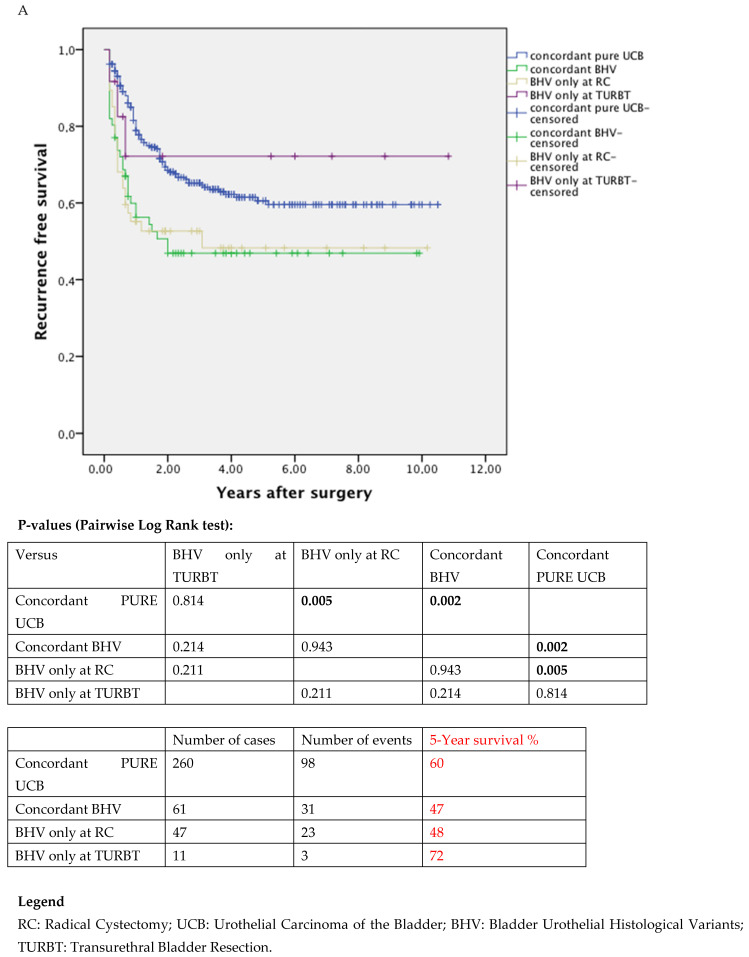
The Kaplan–Meier curve assessing recurrence-free survival rates, (**A**) CSM-free survival (**B**) and OM-free rates (**C**) stratified for histology between TURBT and RC.

**Table 1 jcm-11-00550-t001:** Clinical–pathological characteristics of patients after RC and TURBT according to the TNM classification (VIII edition).

**Variables at Radical Cystectomy**	**N (%):**
**Age at surgery (years, IQR):**	71 (40–90)
**Gender, n%:**	
Male	340 (82.9)
Female	70 (17.1)
**Pathologic tumour stage, (n%):**	
T0	31 (7.5)
Tis	82 (20)
Ta	6 (1.5)
T1	56 (13.6)
T2	71 (17.4)
T3	117 (28.5)
T4	47 (11.5)
**Concomitant Cis, n%:**	
Yes	161 (39)
No	249 (61)
**Pathologic nodal stage, n (%):**	
N0	293 (71.5)
N+	117 (28.5)
**LVI, n (%):**	
Present	122 (29.8)
Absent	288 (70.2)
**Adjuvant Chemotherapy:**	
Yes	42 (10.2)
No	368 (89.8)
**Postoperative follow-up, (median in months, range):**	36 (3–116)
**Time elapsed until first recurrence (median in months, range):**	29 (2–106)
**Variables at TURBT**	**N (%):**
**Gender, n%:**	
Male	340 (82.9)
Female	70 (17.1)
**Tumour stage at last TURBT, (n%):**	
Pure Tis	
T1	24 (5.8)
T ≥ 2	108 (26.4)
	278 (67.8)
**Concomitant CIS, n (%):**	
Yes	119 (29)
No	291 (71)
**Number of TURBT previous to RC, median (range):**	2 (1–5)
**Radical resection, n (%):**	233 (57)
**Focality:**	
Monofocal	315 (77)
Multifocal	95 (23)
**BHV at TURBT, n (%)**	73 (18)

RC: Radical cystectomy; IQR: Interquartile range; LVI: lymphovascular invasion; BHV: Bladder histological variant; TURBT: Transurethral bladder cancer.

**Table 2 jcm-11-00550-t002:** Presence of BHV at pathologic examination in transurethral resection and subsequent radical cystectomy. WHO 2004 (third edition) classification for tumours of the urothelial tract.

**379 Patients**	**TURBT (n, %)**	**RC (n, %)**
Pure UCB	307 (81.0)	272 (71.7)
UCB with Squamous differentiation	39 (10.2)	61 (16.0)
Micropapillary	11 (2.9)	15 (3.9)
Small cell	8 (2.2)	9 (2.4)
Sarcomatoid	5 (1.3)	6 (1.6)
UCB with Glandular differentiation	4 (1.1)	8 (2.1)
Nested	4 (1.1)	6 (1.6)
UCB with Trophoblastic differentiation	1 (0.2)	1 (0.2)
Giant cell	0 (0.0)	2 (0.5)

BHV: Bladder histological variant; TURBT: Transurethral bladder cancer; RC: Radical cystectomy; UCB: Urothelial cancer of the bladder.

**Table 3 jcm-11-00550-t003:** Univariable and Multivariable logistic regression predicting extra-vesical disease (T ≥ 3) (**A**), nodal metastasis at RC (**B**) and positive surgical margins (**C**).

**(A) Extra-vesical disease at RC (T ≥ 3)**				
	**Univariable**		**Multivariable**	
	**OR (IC 95%)**	***p*-Value**	**OR (IC 95%)**	***p*-Value**
Age	1.02 (0.87–1.15)	0.06	1.02 (0.99–1.06)	0.10
Gender (M vs. F)	1.2 (0.58–2.32)	0.23	1.2 (0.61–2.28)	0.60
Presence of BHV at TURBT	3.2 (1.22–5.30)	<0.001	2.2 (1.17–4.30)	0.020
LVI	6.5 (2.05–13.45)	<0.001	7.1 (4.02–12.66)	<0.001
Nodal stage: N+ vs. N0	4.6 (1.13–9.55)	<0.001	4.8 (2.68–8.60)	<0.001
**(B) Nodal metastasis at RC**				
Age	1.2 (0.95–1.12)	0.06	(0.97–1.03)	0.90
Gender (M vs. F)	1.3 (0.54–2.026)	0.23	1.0 (0.49–2.05)	0.90
Presence of BHV at TURBT	2.6 (1.03–5.27)	<0.001	1.6 (0.83–3.13)	0.20
LVI	8.5 (4.17–13.60)	<0.001	6.6 (3.75–11.70)	<0.001
Pathologic stage: T3–4 vs. T0–2	3.7 (1.52–7.33)	<0.001	4.8 (2.65–8.53)	<0.001
**(C) Positive Surgical Margins**				
Age	1.0 (1.00–1.09)	0.01	1.02 (0.98–1.07)	0.14
Gender (M vs. F)	1.5 (0.75–3.26)	0.23	1.4 (0.66–3.25)	0.30
Presence of BHV at TURBT	0.8 (0.38–1.67)	0.56	1.3 (0.60– 3.136)	0.41
LVI	6.5 (3.35–12.42)	<0.001	3.1 (1.40–7.03)	0.001
Pathologic stage: T3–4 vs. T0–2	6.2 (3.07–12.62)	<0.001	2.9 (1.25–6.88)	0.014
Nodal stage: N+ vs. N0	3.7 (2.03–6.99)	<0.001	1.2 (0.61–2.73)	0.53

RC: Radical cystectomy; BHV: Bladder histological variant; TURBT: Transurethral bladder cancer; LVI: lymphovascular invasion.

**Table 4 jcm-11-00550-t004:** Univariable and Multivariable Cox regression predicting disease recurrence, CSM and OM in patients treated with RC based on TURBT features.

Variables	Recurrence				CSM				OM			
	Univariable		Multivariable		Univariable		Multivariable		Univariable		Multivariable	
	HR (IC 95%)	*p*-Value	HR (IC 95%)	*p*-Value	HR (IC 95%)	*p*-Value	HR (IC 95%)	*p*-Value	HR (IC 95%)	*p*-Value	HR (IC 95%)	*p*-Value
Age at surgery	1.1 (0.88–1.17)	0.912	NS	NS	0.9 (0.89–1.16)	0.549	NS	NS	1.2 (1.08–3.16)	<0.001	1.00 (0.98–1.01)	<0.001
Gender (ref: Female)	2.4 (1.59–3.50)	<0.001	2.55 (1.72–3.79)	<0.001	1.4 (1.01–2.59)	0.008	NS	NS	1.5 (1.03–2.85)	0.01	1.55 (1.22–3.79)	0.014
Pathologic characteristics at RC:												
Presence of BHV	2.6 (1.21–5.25)	<0.001	2.78 (1.15–3.44)	<0.001	2.5 (1.39–3.21)	<0.001	1.67 (1.12–2.33)	0.02	1.6 (1.19–4.51)	0.002	1.74 (1.17–2.57)	0.006
Pathologic tumour stage: T0–2 vs. T3–4	2.8 (1.84–5.69)	<0.001	2.26 (1.52–4.81)	0.001	2.8 (1.84–5.19)	<0.001	2.52 (1.66–3.21)	<0.001	1.5 (1.11–2.88)	<0.001	1.82 (1.27–2.61)	0.001
Margins: positive vs. negative	4.7 (1.11–8.36)	<0.001	1.96 (1.29–3.44)	0.001	1.6 (1.14–3.17)	<0.001	1.57 (1.08–2.27)	0.03	1.6 (1.08–2.25)	<0.001	1.73 (1.18–2.53)	0.004
Pathologic nodal stage: N+ vs. N0	2.8 (1.85–5.87)	<0.001	2.41 (1.65–3.57)	<0.001	2.9 (1.17–4.26)	<0.001	1.65 (1.12–4.57)	0.01	2.9 (1.84–4.98)	<.001	2.30 (1.63–3.26)	<0.001
LVI	3.2 (1.99–5.65)	<0.001	2.62 (1.74–4.22)	<0.001	1.9 (1.12–3.33)	<0.001	1.49 (0.97–1.98)	0.06	2.8 (1.85–4.63)	<0.001	1.70 (1.17–2.47)	0.005
Adjuvant Chemotherapy	0.7 (0.25–0.98)	<0.001	NS	NS	0.5 (0.22–0.95)	<0.001	NS	NS	0.5 (0.26–0.98)	<0.001	NS	NS
Pathologic characteristics at TURBT:												
Single vs. Multiple episodes	2.9 (1.34–5.89)	0.040	NS	NS	2.2 (0.89–4.25)	0.35	NS	NS	1.2 (0.66–2.54)	0.20	NS	NS
Presence of BHV	1.5 (1.07–2.29)	0.01	1.52 (1.01–2.15)	0.02	0.6 (0.35–1.25)	0.08	NS	NS	1.6 (1.19–2.51)	0.04	NS	NS
T Stage: T2 vs. T < 2	1.3 (0.89–2.95)	0.51	NS	NS	2.8 (1.52–5.25)	0.05	NS	NS	0.8 (0.55–2.22)	0.38	NS	NS
Concomitant CIS	2.7 (1.27–4.85)	0.04	NS	NS	2.5 (1.28–6.01)	0.02	NS	NS	1.8 (0.94–3.77)	0.09	NS	NS
Radical resection	0.8 (0.52–2.12)	0.18	NS	NS	0.8 (0.29–2.15)	0.09	NS	NS	1.2 (0.620–2.84)	0.25	NS	NS
Focal vs. multifocal tumour	2.5 (0.89–5.29)	0.06	NS	NS	1.8 (0.73–3.05)	0.80	NS	NS	1.6 (0.75–3.55)	0.09	NS	NS

CSM and OM: Cancer specific and overall mortality; RC: Radical cystectomy; BHV: Bladder histological variant; TURBT: Transurethral bladder cancer; CIS: carcinoma in situ.

## Data Availability

Data are available at the institutional bladder cancer database of ASST Ospedale Papa Giovanni XXIII, Bergamo.

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
