# Peer review of "The Accuracy of Transurethral Bladder Resection in Detecting Bladder Cancer Histological Variants and Their Prognostic Value at Radical Cystectomy"

_jcm, 2022, doi:10.3390/jcm11030550_

Round 1

Reviewer 1 Report

This study has focused in a topic of signiifcant interest. The paper is well-written and the results are clearly reported. However some questions need to be clarified or corrected:

  1. Were patients with non-muscle invasive bladder cancer (NMIBC) included in this study? In table 1 baseline clinical staging before cystectomy should be detailed. Likewise in tumor stage results also in table 1,  should be clarified which patients were classified as pT0, pT1/Tis/pTa and pT2. Patients with NMIBC at TURBT and/or cystectomy probably should be analyzed separately of MIBC
  2. The main discordant results between histological fundings in TURBT and cystectomy  were in patients with squamous variant. Can the authors explain  these results?
  3. Patients with squamous variant were the largest group as in other studies. I think it would be interesting to report the results and prognostic value specifically for this group of patients
  4. As the authors mention in the conclusions it would interesting to analyze the pronostic value according the percentage of histological variant
  5. In spite of a signifcant percentage of patients had locally advanced tumors at cystectomy only 10 % of patients received adjuvant chemotherapy . Can the authors explined these findings? To analyze the relationship between presence  of HV and the efficacy of  adjuvant chemotherapy could be an aspect of interest in this study
  6. There is an excess number of tables and figures in the manuscript . It would be advisable to focus  on the most relevant results.

Reviewer 2 Report

The authors reported about the detection rate and concordance of variants of urothelial carcinoma diagnosis between TURB and radical cystectomy specimen in a significant retrospective series of 410 patients. They also explored the prognostic value of variants either at TURB and/or radical cystectomy. The work is not innovative but the revision of WHO classification in 2016 and the size of this contemporary series might provide some interest for the bladder cancer medical community.

Some majors concerns should be addressed:

1) a better definition of the variants' diagnosis criteria should be provided: the WHO classification book should be referenced, the nomenclature used within the text and tables is simplified : as an example, squamous instead of squamous divergence, small cell is not considered as a variant but as a distinct histological type, etc. These simplified designations could be used but it should be stated somewhere that they are not the official names and/or the correspondence with the WHO classification should be explicited.

Likewise, the variation for the % of variants reported within the literature is partly related to different thresholds used to diagnose a variant : the authors should explicit what was this threshold (1%? 5%? 10%? more?). It is also important to explain what was decided and reported when more than one variant was present.

2) The number of pT0 should be explicited. Related to this, it is not possible (as it appears in methods section) to infer for pT0 cases the same status for the cystectomy as for the matched TURB(variant vs pure urothelial carcinoma) : these pT0 cases must be discarded for the concordance study between TURB and cystectomy, and they can be kept for the prognosis study.

3) The statistical plan is not appropriate : there are too many statisticals tests with no correction. The likelihood of false associations is too high. The authors should select some hypothesis to be tested and built tables accordingly. Likewise, the presence of variants should be one cofactor included within the multivariate analysis rather than comparing statistical analyses performed separately for pure urothelial carcinoma and variants. A statistician should be actually included within these analyses.

4) in the discussion, the authors should not be so assertive when commenting the response to NAC and/or Immunotherapy according to the presence of variants. Though this a very hot question, up to now the level of evidences for the reported studies is not sufficient to make any recommendation at clinical level. This supports the interest of multicentric and prospective studies.

Reviewer 3 Report

The data about prognosis and treatment of the variant histology of BCa are still immature and assessed mostly in cystectomy patients. Although most of the histological variants show similar oncological outcomes after radical cystectomy (RC), some subtypes including signet ring cell, spindle cell, and neuroendocrine tumours showed inferior survival compared with pure urothelial bladder cancer.

Here, authors showed that TURBT showed low sensitivity and specificity in detecting BHV. Furthermore, presence of BHV at RC served as an independent predictor of CSM.

There are minor but several language corrections necessary, e.g. ‘at the time of analysis, 66 out of 116 patients with BHV (57%) and 133/284 with pure 146 UCB (47%) were dead (line 146)’ or ‘We retrospectively analyzed data from patients treated with TUR who underwent to RC excluding those treated with NAC to evaluate if diagnosis of BHV can be accurately made with TUR alone (line 345)’.

Table 3 describes predicting extra-vesical disease (T≥3) and nodal metastasis at RC not the fact that presence of BHV at TURBT was associated with pathologic upstage (T>2, p=0.020), albeit not being a predictor for nodal metastasis (table 3) (line 145).

In fig 1 there is OS of 5 yr presented while only 36 mths follow-up are reported (tab 1), while in Fig 1, a, b, c the line goes up to 12 yrs.

In line 156 authors claimed that Oncological outcomes in patients with a BHV only at TURBT compared to those who present BHV at RC (figure 5-7) were not statistically different. Is it correct as figures may show sth on the contrary? please change the time line as it goes up to 12 years.

There are Citations missing 335-7.

Lines 119 and 120 should be removed.

Table 3 and 4 and 5: please correct univariate and multivariate.

Round 2

Reviewer 1 Report

No

Reviewer 2 Report

The authors adressed most of the issues. However, it would be appropriate to include within the Material and Methods section these precisions :

"For urothelial cancer, urothelial carcinoma variants or divergence and grade were classified according to the WHO/ISUP 97 2016 grading system [7]."

and (as specificied in their answers to the reviewer)

"The threshold used for variants was 5% and no cases were described as having more than one variant in the same specimen."

Author Response

We thank the reviewer.

We revised English language and style.

As suggested we added in the Material and Methods section the following sentence: "The threshold used for variants was 5% and no cases were described as having more than one variant in the same specimen."

We previously reported in the Material and Methods section the pathological classification used as follow:

"Pattern of classification was based predominantly on morphological features on haematoxylin and eosin-coloured sections, following the WHO 2004 (third edition) classification for tumours of the urothelial tract [12]." 

"All surgical specimens were processed, described and reviewed by a team dedicated uropathologists. For urothelial cancer, grade was classified according to the WHO/ISUP 2016 grading system [7]"

Reviewer 3 Report

The authors have responded to all the points in the review. 

Author Response

We thank the reviewer.